# What Do In Vitro and In Vivo Models Tell Us about Anisakiasis? New Tools Still to Be Explored

**DOI:** 10.3390/pathogens11030285

**Published:** 2022-02-23

**Authors:** Serena Cavallero, Ilaria Bellini, Antonella Pizzarelli, Stefano D’Amelio

**Affiliations:** Department of Public Health and Infectious Diseases, Sapienza University of Rome, 00185 Rome, Italy; ilaria.bellini@uniroma1.it (I.B.); antonella.pizzarelli@uniroma1.it (A.P.); stefano.damelio@uniroma1.it (S.D.)

**Keywords:** anisakiasis, in vitro model, in vivo model, immune response, host–parasite interplay

## Abstract

Anisakiasis is a zoonosis caused by the ingestion of raw or undercooked seafood infected with third-stage larvae (L3) of the marine nematode *Anisakis*. Based on L3 localization in human accidental hosts, gastric, intestinal or ectopic (extra-gastrointestinal) anisakiasis can occur, in association with mild to severe symptoms of an allergic nature. Given the increasing consumption of fish worldwide, the European Food Safety Authority declared *Anisakis* as an emerging pathogen. Despite its importance for public health and economy, the scientific literature is largely characterized by taxonomic, systematic and ecological studies, while investigations on clinical aspects, such as the inflammatory and immune response during anisakiasis, using a proper model that simulates the niche of infection are still very scarce. The aims of this review are to describe the clinical features of anisakiasis, to report the main evidence from the in vivo and in vitro studies carried out to date, highlighting limitations, and to propose future perspectives in the study field of anisakiasis.

## 1. Introduction

Fish-borne parasitic helminthiasis caused by trematodes, cestodes and nematodes has recently emerged as a major food safety concern, which causes economic impact and significant public health concern. In fact, fishery and aquaculture sectors are growing in global production and consumption [1], and the growing popularity of eating raw or undercooked fish, such as sushi, sashimi, carpaccio or fish tartare, has led to a consequent increase in risk of human exposure to the infective stages of fish-borne helminths [2]. Among these, *Anisakis* spp. (Nematoda: Anisakidae) is the etiological agent of the gastrointestinal zoonotic disease known as anisakiasis. The genus *Anisakis* includes nine species distributed worldwide, and two of them are mainly associated with human infections [3]: *Anisakis simplex* sensu stricto, occurring in the subarctic and temperate waters of the northern hemisphere, and *Anisakis pegreffii*, distributed in the Mediterranean Sea and widespread in the Austral region.

*Anisakis* spp. are gastrointestinal ascaridoid nematodes that infect the digestive system of marine vertebrates. They have a complex life cycle in which developmental stages are transmitted along different levels of the trophic web. Definitive hosts are marine mammals, which release eggs with feces and, after larval molting from stage 1 to stage 3 (L1, L2, L3), they are ingested by crustacean hosts. Then, the infective stage L3 reaches other suitable hosts (paratenic hosts), such as fishes and squids, based on the prey–predator relationship [4]. Humans become accidental hosts after the ingestion of raw or undercooked fish containing L3 larvae. 

The first case of anisakiasis was diagnosed in the Netherlands in the 1960s [5], and, to date, more than 20,000 cases have been reported, of which 90% were in Japan (2000/3000 every year), most likely due to the traditional, local dietary habits. In Europe, estimates range from 20 to 500 cases/year [6,7] according to hospital discharge records and published case reports, and a quantitative risk assessment indicated a risk of anisakiasis in between 7700 and 8320 cases annually in Spain [8]. Considering the intrinsic limitations of currently available diagnostic tools, the heterogeneity of symptoms and the paucity of epidemiological data, the global prevalence of anisakiasis is likely to be severely underestimated [4]. Moreover, *Anisakis* is the only fish parasite able to trigger an allergic reaction in humans by sensitization occurring via infection by live larvae [2]. Allergic reactions may arise even with cooked fish, given the thermostability of some *Anisakis* allergens [9].

An additional aspect not completely understood is the helminths’ ability to induce tumorigenic lesion. Helminths are extremely successful parasites in establishing chronic infections if untreated, causing serious consequences and complications in affected patients [10,11]. Infections with the liver flukes *Opisthorchis viverrini* and *Clonorchis sinensis* and the blood fluke *Schistosoma haematobium* are recognized as definitive causes of cancer [12], but the tumorigenic potential of cestodes and nematodes is not completely understood. In this scenario, reports of tumor co-occurrence with *Anisakis* infective, third-stage larvae (L3) are increasing [13,14,15,16,17], and L3 behavior sometimes mimics metastatic tumors [18,19,20,21]. Nonetheless, exposure to *Anisakis* is suggested as a risk factor for the development of gastrointestinal tumors, and an earlier *Anisakis* infection might be a risk factor for the development of stomach or colon cancer [22]. Unfortunately, few studies aiming to investigate the molecular mechanisms of anisakiasis in humans in detail have been carried out so far, and existing in vitro and in vivo models do not always sufficiently account for the human and disease background. Moreover, some limitations can be identified in literature studies, in terms of reliability of assays and data comparability. In this review, we aim to describe clinical features and immunological aspects related to anisakiasis, including the main evidence from the in vitro and in vivo studies carried out so far on the field, highlighting neglected aspects and focusing on potentially suitable innovative approaches and frontiers to be explored.

## 2. Anisakiasis

Different anisakid nematodes may cause anisakidosis, most frequently species belonging to the *Anisakis* and *Pseudoterranova* genera or, very rarely, to the genus *Contracaecum*. Hereafter, the term anisakiasis will be used to refer to the disease, as most of the evidences reported here were based on *Anisakis* spp. 

Although larval nematodes cannot reach the adult stage in humans, L3 is able to perforate the gastrointestinal tract, causing severe pathological consequences. Acute or moderate, non-specific clinical manifestations are commonly reported. In addition, the onset of serious allergic reactions, such as asthma, hives or urticaria, dermatitis and anaphylactic shock, can be observed [23]. The disease can be classified as acute and moderate by the entity of symptoms and into gastric anisakiasis (GA) and intestinal anisakiasis (IA) or extra-gastrointestinal anisakiasis (ectopic), depending on larval localization. Other forms of anisakiasis are recognized, such as the gastro-allergic anisakiasis (GAA) caused by an IgE-mediated allergic reaction and an asymptomatic form observed in seropositive cases [24]. In some cases, larvae were found in the gastroesophageal junction [25] and in the ileocecal region [26].

In acute forms of GA, early symptoms appear 4–6 h after the ingestion of infected food, causing nausea, vomiting and epigastric pain. Moderate forms of GA are characterized by appetite loss, epigastralgia and the occurrence of gastric pseudotumors. On the contrary, acute signs appear later in IA (around 7 days after eating infected seafood), with abdominal pain, nausea, vomiting, fever, diarrhea, fecal occult blood, leukocytosis and, rarely, eosinophilia [27]. Several rare, extra-gastrointestinal localizations were documented in the abdominal cavity [28], mesenteries and omentum [29] and liver [30], among many other locations. GAA is the most common clinical form, showing the abovementioned gastric symptoms and signs combined with allergic reactions [23]. Chronic gastrointestinal forms are associated with appetite loss, epigastralgia and possible granuloma formation [31]. Due to the heterogeneity of symptoms, the diagnosis of anisakiasis is complicated. Three diagnostic approaches are commonly used: endoscopic examination, which may show a gastric or duodenal ulcer with the worm in close proximity; X-ray examinations with contrast media, which can reveal mucosal inflammation and a narrowing of the intestinal lumen; and hematochemical assay with particular attention to specific and total IgE antibodies [32]. Additional imaging techniques, such as CT or RMN, may be of diagnostic support. A good medical anamnesis is essential before carrying out invasive clinical tests. An endoscopic diagnostic approach may be also curative, in the case of larval removal, and a colonoscopy diagnostic approach should be considered, together with gastric investigation. The anthelmintic drugs used are albendazole and ivermectin, about which, however, there are conflicting opinions regarding their real efficacy in vivo [33]. The effectiveness of alternative drugs, such as gastric acid secretion inhibitors and gastric mucosal protectants, are under evaluation in Wistar rats model [34]. Regarding this, EFSA expressed its opinion in 2010, stating that, having not identified specific and targeted pharmacological treatments for the effective killing of the parasite in vivo, the most effective treatment remains prevention, as the eradication of such nematodes in natural settings is not desirable [2].

## 3. Features of Immune Response to Helminthiasis and Anisakiasis

The gastrointestinal helminths of public health concern usually use humans as natural, final hosts, while the natural life cycle of anisakids includes several aquatic organisms, and humans are accidental hosts not accounted for in the natural evolution processes, such as co-evolution and/or co-adaptation, that have occurred over time between hosts and parasites [35]. Despite that, only one L3 is sufficient to cause clinical outcomes in humans. A few hours after ingestion, the L3 reaches the gastrointestinal tract. Through a combination of mechanical tissue disruption and release of secrete/excrete (E/S) factors, such as soluble, potent proteolytic enzymes able to degrade the extracellular matrix [36], together with the extracellular vesicles, including exosomes, the L3 starts to invade the mucosa and submucosa of the gastrointestinal tract. Audicana et al. [37] described this mechanism in a time-dependent manner, accounting for a progressive process that leads to ulcerous, erosive and granulomatous lesions [37,38]. Helminths, such as *Anisakis*, are macro pathogens, a condition that prevents them from being quickly captured by phagocytic cells. Therefore, immunomodulating messengers released by immune cells and the parasite itself play a pivotal role in host–parasite interactions, contributing to the onset of an inflammatory microenvironment able to induce a specific Th2 response [39,40]. The epithelial barrier, macrophages (MØ) and dendritic cells (DCs) represent the first line of the human innate immune response. A further pivotal element of the innate immune response is the receptors and, among them, toll-like receptors (TLRs) are the most investigated in several helminthic infections [41]. In particular, DCs identify *Schistosoma mansoni* lipid antigens containing phosphatidyl serine through TLR2, while glycoprotein ES-62 of filarial nematodes is recognized through TLR4 [42]. Despite their relevance, information about TLRs in anisakiasis is still poor and not clear. Usually, the activation of TLRs leads to downstream intracellular signaling. The MAPK pathway is one of the main signaling cascades associated with TLRs stimulation, and it is composed of the extracellular, signal-related kinases 1 and 2 (ERK 1/2), p38 MAPK and c-jun NH2-terminal kinase (JNK). ERK 1/2 usually induces Th2 activation during infections due to extracellular pathogens [43], while P38 and JNK are mostly involved in the Th1 responses commonly reported for intracellular pathogens [44]. Once activated, these molecules induce gene expression, DC maturation and cytokine production via the phosphorylation of transcription factors, such as activating protein 1 (AP-1) and nuclear factor-κB (NF-κB) [45]. The cytokine and chemokine milieu derived from the innate immune cells activation results in the release of specific, pro-inflammatory products, such as thymic stromal lymphopoietin (TSLP), eotaxin and interleukins, such as IL-25, IL-4, IL-5, IL-13, IL-9 and IL-33 [40], that contribute to DC activation and type 2 response initiation. It was demonstrated that mice deficient in IL-25 are susceptible to infection with the rodent whipworm *Trichuris muris*, and mice with a dysregulation in TSLP production are unable to induce Th2-driven worm expulsion [46]. The tissue damage and the release of inflammatory products allow the recruitment of competent cells, such as neutrophils, eosinophils and basophils, among several others, in the site of damage. Actually, one of the main signatures of local lesions produced by *Anisakis* larvae is the presence of an appreciable eosinophilic infiltration in the tissues surrounding the parasite [37]. At this point, mature DCs migrate to lymph nodes, priming T naïve cells and giving arise to a Th2 clonal proliferation. The triggering of the Th2 response induced by intestinal nematodes leads to a stereotyped signal cascade of effector mechanisms, including immunoglobulin E isotype-switched B-cell responses, increased permeability, epithelial cell turnover, smooth muscle contractility, mucus production, eosinophilia and mastocytosis, with consequent parasite expulsion [47]. To avoid this, helminths have devised different strategies to manipulate the host immune system. The alarmin release inhibitor E/S product HpARI released by murine intestinal nematode *Heligmosomoides polygyrus* suppresses the IL-33 ablating Th2 response [48,49]. Again, asu-miR-5361-5p, a miRNA observed in adult pig roundworm *Ascaris suum* exosome content, downregulates CD80, a receptor involved in the induction of T-cell proliferation and cytokine production [50]. The suppression of the host immune response and tolerance can lead to a parasite persistence and chronic inflammation. During chronic anisakiasis, the persistent stimulus of the L3 amplifies the inflammatory state and continuously attracts eosinophils, as well as neutrophils, lymphocytes, monocytes and fibroblasts, recruited to deposit connective tissue to form a granuloma [38]. The main role of granuloma is to protect the host from pathogens and/or persistent irritants and, after larval death, *Anisakis* remains are broken down, in time, becoming unrecognizable. This phenomenon contributes to complication of the diagnosis; thus, *Anisakis*-induced granulomas have been misdiagnosed for tumors in the past but gradually disappear in infected patients, leading to the term “vanishing tumors”.

## 4. In Vitro Models

In vitro models were used to investigate the pathogenic potential of L3 in relation to cellular oxidative stress and immunological and inflammatory responses during anisakiasis. 

The ability to penetrate tissue of natural and/or accidental hosts or to survive in gastric juice may be indicative of L3 pathogenicity, and a few studies aimed to explore such abilities in *Anisakis* spp. and reveal any difference in the pathogenic potential among species. Four independent studies inferred a natural, higher pathogenicity for *A. simplex* s.s., given its higher penetration rates in fish fillet and in agar and its survival in artificial gastric juice, and a lower but still present pathogenic ability for *A. pegreffii* [3,51,52,53]. 

Pathogenic potential may also be related to the degree of host cellular response after exposure to live parasitic developmental stages or parasite-derived material such as crude extract (CE) or E/S material. However, many parasitic nematodes cannot be easily cultured and existing in vitro and in vivo models do not sufficiently recount the human background that reflects the natural niche of infection. However, some attempts have been made to explore such aspects using human-derived cellular in vitro cultures, and the main evidence derived from these attempts is summarized in Table 1.

Studies using non-immune cells as human fibroblasts may be of support in the understanding of aspects related to larval migratory behavior and the disruption of extracellular matrix, given that fibroblast synthesizes connective tissue. By treating cells with CE and E/S, the upregulation of ROS, p53, c-jun, c-fos and Hsp70, leading to inflammation and apoptosis inhibition (activation of cellular proliferation), was reported [54]. 

The role of immune cells and the effect of two types of *Anisakis* products (live L3 and CE) on immune cell maturation were investigated [55] using human dendritic cells (DCs). Human DCs are fundamental in antigen presentation as they induce naïve T-cell activation and effector differentiation and, subsequently, activate the shift from innate to adaptive immune response. After treating immature and mature DCs with L3 and CE, a different modulation in the DC secretion of cytokines and chemokines was observed. While both triggered an upregulation of IL6, only live L3 treatment was associated with significant downregulation of IL10, CXCL10, CCL4 and ICAM and upregulation of CCL3, while a mild, non-significant decreasing trend was observed in CE-treated cells. Both conditions impacted cellular viability and maturation by reducing the expression of molecules, such as HLA-DR, CD86, CD83 and CCR7, involved in antigen presentation and migration and by increasing ROS level. Autologous CD4+ T cells, stimulated with DCs exposed to *Anisakis* L3 or CE, failed to produce IL-4, IL17 and IL10, suggesting a less reactive phenotype of DC not sufficient to drive a Th2/Th17 immune response.

The human epithelial colorectal adenocarcinoma cells Caco-2 have been widely used as a model of the intestinal epithelial barrier. Their ability to differentiate into a monolayer of cells with many properties typical of the small intestine makes this model particularly advantageous for studying intestinal parasitic infections and inflammatory processes. Using this model, modulatory activity exerted by *Anisakis* parasitic products, such as upregulation of oxidative stress, inhibition of apoptosis-related biomarkers and inflammatory induction, were reported [56]. In fact, CEs induced a marked decrease in caspase-3 activation and marked COX-2 expression, suggesting that exposure to *Anisakis* may affect various cellular pathways crucial not only in the inflammatory process, but also in proliferation and death. Recently, the same cellular model was used to study the effects of *A. simplex* CE on the intestinal integrity and permeability in relation to the Ani s 4 allergen [57]. The authors observed a decrease in trans-epithelial electrical resistance after CE treatment not explained by cytotoxicity or by parasitic protease activity. The altered integrity and barrier function were associated with increased ROS production and changes in tight junction (TJ) protein localization. Moreover, the thermostable Ani s 4 antigen resistant to pepsin digestion was transported to the basolateral side crossing the intestinal barrier, as reported for other allergens [58]. 

## 5. In Vivo Models 

Animal models to explore the pathogenic behavior of infective larvae in anisakiasis are still of limited use. So far, aspects related to immune response, inflammation, allergy and tumors have barely been investigated; three models have been used, BALB/c mice, Wistar rats and Sprague Dawley rats.

BALB/c mice were the most used animal model to explore allergic and lesion-related aspects of anisakidosis. The presence of anti-IgG sero-reactivity to somatic and E/S antigens was demonstrated in BALB/c mice following orogastric inoculation of *Anisakis* spp. but was less pronounced with *Contracaecum* spp. [59].

Routes of allergic sensitization to *Anisakis* were explored in two independent studies using interleukin-4 receptor alpha-deficient BALB/c mice [60,61]. First evidence suggested that *Anisakis* proteins induce allergic reactions in sensitized mice by IL-4/IL-13-mediated mechanisms [60]. This was confirmed by further experiments in relation to different kinds of parasite exposure (intranasal, intraperitoneal and live or heat-killed *Anisakis* larvae). Results revealed that administration and type of parasitic material are potent triggers for allergic sensitization. Depending on the type of administration, allergic airway inflammation was induced either by IL-4/IL-13 or by IFN-gamma. Intranasally sensitized mice failed to trigger *Anisakis*-specific IgE and IgG antibody responses [61]. This may explain food-related allergic episodes and support occupational allergies in humans, even in the absence of specific IgE antibodies. The allergenicity of two important allergens, Ani s 1 and Ani s 9, was investigated by repeated intranasal administration of the allergens [62], revealing a high ability to induce Th2 (IL-4, IL-5, IL-13 and IL-25) and Th17 (IL-6, IL-17, IL-25 and CXCL1) responses in the lung. Later, the allergenic potential of *Anisakis typica* was tested by oral administration of crude extract to BALB/c mice [63]. Results showed the elicitation of patterns not entirely related to food antigen allergy, showing Th1/Th2-related cytokines with mixed pattern tending to inflammation. In fact, some inflammatory-related markers were triggered (CD11c+IL-6+, CD4+IFN-γ+, CD4+IL-4+, CD4+IL-5+), indicating that *A. typica* challenge induces DC maturation to secrete IL-6 as a pro-inflammatory cytokine. At the same time, some other markers were reduced (CD4+CD25+CD62L+), suggesting a depletion of naïve Tregs population after one week. Allergenic activity was also investigated for *Pseudoterranova decipiens*, suggesting its residual ability to elicit an IFN-ɣ and Th1/Th2 response [64]. A mice model was used to also explore severe allergic reactions, such as anaphylaxis induced by *A. simplex*, and specific IgE and IgG, plasma histamine release (only in intravenously challenged not in orally challenged) and IL-10, IFN-gamma, IL-4, IL-13 and IL-5 release were reported [65]. The allergic reaction was also evaluated in Wistar rats with different kinds of allergens (fresh and frozen L3, heat-treated, intraperitoneal sensitizations with CE), causing a significant rise in Th-related cytokines (IFN-γ, TNF-α, IL-5, IL17 and total IgE) [66].

Hypothetically, all anisakids may be infective or allergenic for humans, but not all species have been reported in association with human cases. Whether this is due to epizootiology aspects, such as occurrence of certain *Anisakis* spp., in largely consumed fish or due to intrinsic parasitic features related to infectivity is still unknown. These possible differences in pathogenic larval behavior in relation to lesions, attachment to the gastrointestinal wall, tissue migration and tumorigenic potential were investigated mostly using Sprague Dawley rats. In fact, histopathological characterization of lesions showed the presence of neutrophils and macrophages and extensive submucosal hemorrhages, necrotic areas and increased vascular permeability in the stomach and intestine [67].

*Anisakis* CE induced increases in cell proliferation and decreases in apoptosis, suggesting a residual ability to induce tumors in accidental human hosts [68]. Host-circulating miRNAs were analyzed, and significant differences in expression were reported. The mmu-miR-1b-5p and mmu-miR-10b-5p (a cancer-related miRNA) and rno-miR-218a-5p and mmu-miR-224-5p were significantly decreased in sera from the treated rats, while rno-miR-125a-3p and rno-miR-200c-3p were increased. *Anisakis* CE was able to induce changes both in epithelial cells in vitro and in an animal model. Despite the great relevance for public health and the large occurrence of L3 in fishes intended for human consumption, the tumorigenic potential of anisakids is still largely neglected. On the contrary, several attempts have been made to explore the pathogenic potential of different species belonging to the Anisakidae family. *A. simplex* s.s. and *A. pegreffii* [3,69], as well as *Anisakis physeteris* and *Anisakis paggiae* [70], showed the ability to reach the stomach, the intestine and the extra-gastrointestinal niche, with slight differences. Matrix metalloproteinases and serine proteases (especially plasmin), produced to degrade the extracellular matrix protein of the gastrointestinal tract and to digest the formation of fibrin during the penetration course, coupled with the mechanical movement of infective larvae, have strong involvement in the penetration of the gastrointestinal tissues of mice [71].

The early phases of the onset of anisakiasis are also interesting steps to investigate, and RNA-seq can assist the description of cellular and molecular mechanisms underlying host responses. An acute, pro-inflammatory host response, predominantly characterized by the activation of a Th17 lineage of effector T-helper lymphocytes, was described [72]. The IL17 signaling pathway associated with the recruitment of neutrophils and an enhanced pro-inflammatory reaction were reported, thus, in turn, stimulating inflammation through necrosis and granuloma formation. 

Aside from the known zoonotic potential of *Anisakis* spp. and *Pseudoterranova* spp., other anisakids may be responsible for human infections. Among them, zoonotic potential was confirmed for *Contracaecum* spp. in relation to human cases reported in Europe, Australia and Japan [73,74,75], even if human cases associated with this genus are rarely reported. In this respect, the pathogenic potential of *C. osculatum*, using the experimental infection of pigs with L3 recovered from Baltic cod, was explored [76]. Pathological reactions included elicitation of eosinophilic granuloma associated with larval penetration of the pigs’ stomach mucosa, with numerous giant cells, lymphocytes, macrophages, granulocytes and fibroblasts observed around the worm. Similar histopathological changes were also reported in experimentally infected CBA/J mice with *Anisakis simplex*, in which aggregations of neutrophils adjacent to the parasites were first observed followed by mature granuloma consisting of eosinophils, fibroblasts and associated collagen, with granulocytes and multinucleate giant cells occupied the host–parasite interface [77].

## 6. Conclusions and Future Perspectives

In recent years, investigation of the pathogenic mechanisms exerted by helminths in humans has acquired even greater importance given that some fish-borne parasites are listed among the Neglected Tropical Diseases, according to WHO. Crucial, biological, parasitic features with fine regulatory functions and the potential ability to ameliorate and mitigate inflammatory diseases and evade host immune response have received much interest given their possible therapeutical application. Although helminthic infections are often underestimated due to their low mortality rates and complicated diagnostic approaches, in vitro and in vivo methods have given appreciable answers to deepen the understanding of helminth-related pathogenesis in humans [78]. Generally, two coexistent opposite responses, Th1 and Th2, seem to be generated by *A. simplex*-derived material, as also observed in other helminthic infections, and the genetic background of the host also plays a role in favor of Th2 or Treg response [79]. However, *Anisakis* is still largely neglected among parasitic helminths. A lack of standardization in the experiments in terms of settings (i.e., amount of L3 used, times of incubations, quantity/dose of starting materials, such as CE, kind of material used, such as live L3, CE, E/S, coadjuvants) emerged as a limiting factor, making comparisons and inferences very difficult. In addition, in vitro models applied to anisakiasis rarely resembled the real niche of infection. To date, only two studies aimed to investigate this zoonosis using the Caco-2 cellular line [56,57]. Human epithelial colorectal adenocarcinoma cells are composed by heterogeneous and immortalized cells that can spontaneously differentiate into a heterogeneous monolayer with a brush border layer, containing cells with slightly different properties, as in the small intestine. Unfortunately, cultivation conditions select subpopulations of cells, meaning that, in a model system with variable properties and results obtained under similar experimental conditions in different laboratories, results may not be directly comparable [80]. In this complex scenario, the in vitro models based on three-dimensional intestinal organoids are the most advanced and powerful tool to study pathogenesis of infectious agents. Organoids are multicellular structures able to better recapitulate the physiologic function of the organ from which they are derived. Intestinal organoids are widely used to study the physiological and pathological intestinal condition, as well as interactions among the host, microbiota and pathogens [81]. To date, few studies are available in parasitology regarding intestinal protozoans [82,83], and only three attempts were performed on helminths such as whipworms, hookworms and *Trichinella* [84,85,86]. The use of intestinal organoids for anisakiasis avoids the existing fragmentation of information, providing a model that better recounts the complexity of human background. Various pitfalls may also result from in vivo models. Despite the medical relevance of the topic, a clear imbalance is evident between studies aiming to explore anisakiasis allergic-related aspects compared to those deciphering the mechanisms of infections in the gastrointestinal tract. Moreover, a lack of standardization in terms of methods of antigen/L3 administration, starting material preparation and times of infection may jeopardize the reliability of outcomes. 

A few in vitro and in vivo studies on anisakiasis revealed well-known features of early cancer development [22,54,56,68]. Given that the relation between chronic inflammation and cancer is well known [87], as well as the tumorigenic potential of some parasitic flatworms [88], further efforts are needed to address this important, unexplored field for *Anisakis*. Nematodes are able to establish long-lived, chronic infections through the release of molecules, broadly referred to as E/S products [40]. Among these products, only antigens and peptidases were considered but, recently, the extracellular vesicles’ (EVs) discovery revived the interest in understanding the potential crucial role of exosomes in host–parasite interplay. The potentiality and content of these vesicles are currently the focus of many studies in medicine, as they are involved in physiological and pathological conditions and cell–cell and host–pathogen interactions. Exosomes are the most studied category of EV since they deliver their cargo in a protected state. Their content includes DNA, proteins and ncRNAs (non-coding RNAs) such as microRNA. miRNAs are endogenous RNAs of 21–23 nt in length, present in all eukaryotes [89], playing relevant roles in every aspect of cell life through post-transcriptional gene regulation. In fact, miRNAs act by promoting degradation and/or translation repression of target messenger RNAs [90]. Considering that miRNAs regulate endogenous genes, pathogens may exploit miRNAs to target host genes to reinforce existing pathways or to downregulate the immune and inflammatory response. This is well known for viruses, and it recently emerged for parasites as well: such a strategy may provide an evolutionary gain because miRNAs, differently from proteins, do not evoke host immune responses. A few miRNAs have been described in nematodes, such as *H. polygirus*, *T. muris* and *A. suum* [50,83,90], with detailed inferences about their predictive targets, usually genes related to immunity. The ability to transfer small RNAs to mammalian cells and to modulate innate immunity was demonstrated for *H. polygirus* in an elegant and detailed study, where administration of the nematode exosomes to mice suppressed type 2 innate responses and eosinophilia induced by the allergen *Alternaria* by suppression of Il33r and Dusp1 [91].

To date, only one study aimed to investigate *Anisakis* exosomes [92], focusing on their release and uptake by host cells and providing their first proteomic characterization, but no data about miRNAs content are available yet. Deepening the knowledge on *Anisakis*-derived extracellular vesicles could reveal undescribed parasite abilities and pathological aspects in host–parasite interactions that, to date, are still obscure, highlighting new, potential diagnostic biomarkers and therapeutic or prevention applications, as demonstrated for other helminths [93,94,95].

## Figures and Tables

**Table 1 pathogens-11-00285-t001:** Summary of in vitro studies aimed to investigate *Anisakis*–human interactions.

*Anisakis* spp.	In Vitro Model	Biological Material	Results	Ref.
*A. pegreffii*	Fibroblast cell line HS-68	ES CE	Both: Upregulation of P53, ROS, cell proliferation. Downregulation of cell viability. ES: increased TNF-α, c-jun, c-fos. CE: increased Hsp70, decreased TNF-α.	[54]
*A. pegreffii*	Human dendritic cells	L3 CE	Both: downregulation of cell viability, ROS, HLA-DR, CD86, CCR7, INF-γ. L3: downregulation of ERK1-2, CXCL10, CCL4, ICAM, CCL3, IL-10. Upregulation of IL-6 and IL-1α. Inaltered NF-KB. CE: upregulation of IL-6 and IL-1α and CCL3. IL-10 inaltered.	[55]
*Anisakis* Type I	Caco-2 cells	CE	Upregulation of COX-2. Downregulation of caspase-3.	[56]
*A. simplex*	Caco-2 cells	CE	Decreased cell monolayer integrity (TEER). Increased permeability with disruption of tight junctions ZO-1, Occludin). Induction of ROS and inaltered cellular respiration. Anis 4 antigen able to transport to the basolateral side crossing the intestinal barrier.	[57]

ES: excrete/secrete; CE: crude extract; L3: live larvae at third stage; Ref.: references.

## Data Availability

Not applicable.

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
