# Peer review of "What Do In Vitro and In Vivo Models Tell Us about Anisakiasis? New Tools Still to Be Explored"

_pathogens, 2022, doi:10.3390/pathogens11030285_

Round 1

Reviewer 1 Report

Manuscript ID: pathogens-1594142

What in-vitro and in-vivo models tell us about anisakiasis? New tools still to be explored

I think it's a very interesting and very important topic in parasites, food borne disease and one-health context nowadays as regards investigations on clinical aspects such as inflammatory and immune response during Anisakiasis using a proper model that simulates the niche of infection.   

The present work aims to describe clinical features of anisakiasis, to report main evidences from in vivo and in vitro studies carried out to date, highlighting limitations and to propose future perspectives in the study field of anisakiasis.

The topic is of interest for the academics and clinicians, the conclusions are of interest and literature is rich and update.

The review is well written and easy to understand by readers. I believe that this review does not need big changes. English & language and style are fine.

Specific suggestions:

Lines 29: stages of fish-borne helminths… please cite

  • Giorgio Smaldone, Rosa Luisa Ambrosio , Raffaele Marrone, Marina Ceruso and Aniello Anastasio, 2020. Anisakis spp. Larvae in Deboned, in-Oil Fillets Made of Anchovies (Engraulis encrasicolus) and Sardines (Sardina pilchardus) Sold in EU Retailers. Animals, 10, 1807

Line 99: remains prevention… I think that prevention nowdays is useful applying risk analysis tools please cite

  • Smaldone, Giorgio; Abollo, Elvira; Marrone, Raffaele; Bernardi, Cristian E. M.; Chirollo, Claudia; Anastasio, Aniello; del Hierro, Santiago P. 2020. Risk-based scoring and genetic identification for anisakids in frozen fish products from Atlantic FAO areas. BMC VETERINARY RESEARCH. 16, (1), 1-12

Line 166: ability for A. pegreffii [2, 43-44].. these in vivo cited studies are confirmed by a published paper. Please cite

  • Cipriani P, Smaldone G , Anastasio A, Acerra V, D’Angelo L, Bellisario B,  Palma G, Nascetti G, Mattiucci S, 2015: Genetic identification and distribution of the larval parasites Anisakis pegreffii and A. simplex (s. s.) in fish tissues of Merluccius merluccius from Tyrrhenian Sea and Spanish Atlantic coast: implications for food safety. International Journal of Food Microbiology, Volume 198, April 02, 2015, Pages 1-8. ISSN: 01681605; DOI: 10.1016/j.ijfoodmicro.2014.11.019.

Author Response

Dear Reviewer, we are grateful for your support and comments. Please find a detailed rebuttal attached, with our replies in blue. We have submitted a revised version of the Review with the changes in red. Sincerely

Reviewer 2 Report

This review, presented by Cavallero and colleagues accurately describes clinical and immunological features of Anisakiasis and provide the main evidences from in-vitro and in-vivo trials. Moreover, the present study highlights the neglected aspects of this sanitary issue and suggesting innovative approaches that should be explored by future researches. Considering the knowledge gap of some clinical aspects of Anisakiasis I recommend publication of the manuscript with only a few minor corrections. Please find more specific comments below.

Line 38-40 Authors state that “in Europe estimates range from 38 20 to 500 cases/year [4-5], but a recent quantitative risk assessment indicated a risk of anisakiasis between 7700 and 8320 cases annually in Spain [6]. Reference 6 was published in 2017 while [4] and [5] in 2018”. In my opinion this statement should rephrased, in this form is misleading for the reader.

213-218 please, provide bibliographic references. This paper is presented as a review therefore I strongly suggest to increase the bibliographic reference throughout the manuscript.

Line 228 Please, correct typing error. In particular, add square bracket to bibliographic reference 51-52.

Line 230 231 which are the further experiments? please add the bibliographic reference. As I commented above, increase the bibliographic references.

Allergic reactions to Anisakis are also reported after the consumption of properly cooked fish products containing Anisakis larvae or allergens which resist to thermic treatment as heat and or freezing. In my opinion, authors should mention this argument and provide the bibliographic references.

Line 358 Please correct the typing error, put the spacing after the square bracket.

Check the reference list carefully again from the beginning. Reference lists are frequently hotbeds of errors. The volume number must be written in italics.

Author Response

(The authors gave the same response as above.)

Reviewer 3 Report

The article by Cavallero et al. summarizes current knowledge on the immune response to Anisakis spp. parasites derived from several in vitro and in vivo studies. Furthermore, the stress several important points for future studies on anisakiasis that should shed new light on underlying mechanisms of host immune response and potentially help develop new diagnostic and therapeutic approaches. Considering the severe underestimation of anisakiasis case and its misdiagnosis and by extent, the disease burden that this zoonosis represents such reviews that will help in future studies are more than welcomed. In fact, to the best of my knowledge, this is the first review of such type pointing out the scarcity of studies on anisakiasis, making this review even more important.

The manuscript is generally well structured with up-to-date literature cited. Nevertheless, the are several major issues that need to be addressed before the manuscript could be accepted for publication, which would also add much to the clarity for the broader audience. Also, minor language editing is needed.

MAJOR COMMENTS

In the introduction section please add a short description of the Anisakis life cycle and indicate that humans are accidental hosts. This is not strictly a parasitological journal and a broader readership is likely not to be familiar with it. Indeed, you mention this later in the manuscript, however it should be added to the introduction, so the rest of the manuscript is easier to follow.

Although this is not a systematic review, I would strongly suggest adding a short section on how the literature was searched. Include here what bibliographic bases were searched, what keyword and/or MeSH terms you used and any additional information that would help the reader repeat the search.

The section on anisakiasis might be a bit confusing to a reader not familiar with the topic. I suggest rewriting it in a way that you first list classification of all clinical entities of anisakiasis and then give a brief description of each entity. I also suggest you use a more recent classification by Moneo et al., 2017 Current Allergy and Asthma Reports who add a fifth group of asymptomatic seropositive patients, which I believe is of high importance for anisakiasis epidemiology.

Section 3. Innate and adaptive immune response in Anisakiasis should be shortened and the title rephrased as it is misleading to the reader. In my opinion, for this kind of review, it is not necessary to describe mechanisms in detail (such as MAPK signalling pathway), but rather mention key cytokines involved and the end result of their activation in eliminating the parasites (e.g. granuloma formation, increased intestinal permeability, weep and sweep response). In this section, among others, you describe immune responses in some well-known and established host-parasite models, with definitive hosts, which are not necessarily attributable to anisakiasis given that the humans are accidental hosts which should also be pointed at the beginning of the section.

In the section on in vivo models of anisakiasis several other studies might be important for the current knowledge on anisakiasis and of importance for the reader. I suggest you consider adding studies by Abdel-Ghaffar et al., 2015 Parasitology Research, Hrabar et al., 2019 PLoS NTDs, Gómez-Mateos et al., 2020 Digestive Diseases and Sciences. The former two dealt also with the immune response of rats to Anisakis infection in respect to different modes of infection and microRNA expression, respectively. The latter one should be considered in light of your comment on developing new therapeutic procedures you mention in the Conclusions sections, as the lack of effective medicinal therapy is also an important aspect of anisakiasis. Although it wasn’t really an anisakiasis model, but rather an in vitro study on parasites themselves, Mladineo et al., 2018, Antimicrobial Agents and Chemotherapy as well as Ogata and Tagishi, 2021 Pharmacology also provided interesting results on the potential use of different medicinal and natural compounds for the treatment of anisakiasis, emphasising the need for more studies on this topic

MINOR COMMENTS

Line 38 Please change food habits to dietary habits.

Line 39-40 Please add that the authors of the paper you refer to here indicate that this is a number of cases requiring medical attention, as anisakiasis could also be asymptomatic, and therefore its incidence higher.

Line 46 What do the authors mean by “successful parasites”? In what sense? Maybe rephrase it to “Helminths are extremely successful parasites in establishing chronic infections…”. The way you put it would suggest establishing chronic infections is just one aspect of their success and the reader is left without indication what the other aspects could be.

Line 71-75 This classification is well known among people working with Anisakis and anisakiasis. However, for a broader audience a proper reference(s) for this statement should be added.

Line 77-82 Same as above, add proper reference(s).

Line 95-96 Same as above, add proper reference(s).

Line 110-113 Same as above, add a proper reference(s).

Line 114-155 To what exactly does “among them” refer to? First, you talk about different cell populations and then you jump to their receptors. It seems to me like there is a sentence linking these two sentences missing.

Line 213-216 As mentioned in the major comments on the Anisakis life cycle please remove this part and move to the Introduction.

Line 216-218 This part is superfluous as you already described it in the introduction.

Line 244 and Line 246 Please remove the + sign. The + sign next to a marker designation is used to describe the immunophenotype of cells and here you are not talking about different populations of cells.

Line 256-259 Please add here that Wistar rats were also used in in vivo studies as the considerable number of studies you refer to have used this strain of rats, not the Sprague-Dawley.

Author Response

(The authors gave the same response as above.)

Round 2

Reviewer 3 Report

The authors have addressed all of the reviewer comments, accepting most of the suggestions which have added considerably to the clarity of the manuscript text. For those suggestions that the authors wished to not accept they have provided valid and strong arguments why to keep the manuscript text as originally written.

Therefore, after the second review round my suggestion is that the manuscript be accepted in the present form.